# Intrinsic hydroquinone-functionalized aggregation-induced emission core shows redox and pH sensitivity

Mengshi Wang[1,7], Yuanheng Wang[2,7], Renjian Hu[1], Jinying Yuan[3], Mei Tian[4], Xiaoyong Zhang[5], Zhigang Shuai [2] & Yen Wei [1,6 ✉]

Aggregation-induced emission (AIE) fluorophores exhibit strong fluorescence in an aggregated state but emit no or weak fluorescence in dilute solutions. This emerging class of AIE optical materials comprise a variety of functionalities. Here an AIE luminescence core, 1-hydroquinol-1,2,2-triphenylethene (HQTPE), has been designed and synthesized. This AIE core is simple but is fundamentally important to chemistry because of its intrinsic redox and pH activities. The incorporation of hydroquinone (HQ) moiety into a common AIE core tetraphenylethene (TPE) yields HQTPE with unique fluorescent properties like nonlinear self-quenching over most other AIE-active fluorophores (AIEgens) so far reported. There are differences of photochemical properties between HQTPE, 1-benzoquinol-1,2,2-triphenylethene (QTPE, the oxidized counterpart) and its anions. Interestingly, as the solution concentration is increased, AIEgen HQTPE shows stronger fluorescence but QTPE exhibits rapid quenching of fluorescence in a nonlinear fashion, which are in agreement with theoretical studies. The fluorescence of HQTPE is also highly dependent on the pH value of media. We have further explored HQTPE as an ultrasensitive redox probe and efficient deoxidizer, which could lead to potential applications in health care, food security, environmental monitoring, optic and electronic devices.

[1] The Key Laboratory of Bioorganic Phosphorus Chemistry and Chemical Biology, Department of Chemistry, Tsinghua University, Beijing, China. [2] MOE Key Laboratory of Organic Optoelectronics and Molecular Engineering, Department of Chemistry, Tsinghua University, Beijing, China. [3] Key Lab of Organic Optoelectronics and Molecular Engineering of Ministry of Education, Department of Chemistry, Tsinghua University, Beijing, China. [4] Department of Nuclear Medicine and Medical PET Center, The Second Hospital of Zhejiang University School of Medicine, Hangzhou, Zhejiang, China. [5] Department of Chemistry, Nanchang University, Nanchang, China. [6] Department of Chemistry, Center for Nanotechnology and Institute of Biomedical Technology, Chung-Yuan Christian University, Chung-Li, Taiwan, China. [7] These authors contributed equally: Mengshi Wang, Yuanheng Wang. ✉email: weiyen@tsinghua.edu.cn

luorescent materials have an important role in modern society because of their extensive applications in chemical and biological sensors, anti-counterfeit, forensic and biomedical imaging, arts and entertainment industries, etc.[1–4]. However, most of traditional fluorophores suffer from aggregation-caused quenching effect[5], which has seriously restricted their applications in concentrated solutions and in solid state. In contrast, aggregation induced emission (AIE) effect has received great attention since the pioneering work by Tang and colleagues[6] in 2001. AIE-active fluorophores (AIEgens) exhibit strong fluorescence in an aggregated state but emits no or weak fluorescence in dilute solutions. Among the AIEgens, tetraphenylethylene (TPE) is the most widely used structure motif[3,7,8].

To expand the applications of AIEgens, it is often practiced to integrate AIE fluorophore and functional groups into an inter-related system[9]. One of the common but important modifications is the introduction of redox or pH-responsive groups[10]. The redox process is one of the most fundamental chemical reactions. Biochemical processes in living organisms[11,12], such as mitochondrial activity, respiratory functions, cancer-related reactive oxygen species (ROS), etc., are always connected to sophisticated cascades of redox reactions. Furthermore, it is almost ubiquitous in industrial production and in prevention of corrosion[13,14]. On the other hand, pH value is one of most basic parameters in solution chemistry, which needs to be measured and controlled in various fields such as agriculture, pharmaceutical, chemical industry, etc.[15,16]. To date, numerous AIEgens with redox or pH responsiveness have been successfully exploited for detection, bioimaging, and quantitative analysis[17]. However, the conventional construction methodology to acquire redox- or pH-responsive AIEgens involves the coupling of typical AIE-active moieties with auxiliary functional groups. For example, acetal, imine, and orthoester commonly serve as pH-sensitive moieties[18–20], whereas ferrocene and disulfide linkage functions serve as as redox-responsive building blocks[10,21]. From these previous works, it is not difficult to find that functional AIEgens in the traditional sense could be prepared through a simple stitching of AIE-active core and functional groups. However, the current functional AIEgens are seriously homogenized and the design idea is not delicate and convenient enough. Except in rare cases[22], most functional AIEgens reported so far were designed through a link between an AIE core and functional groups, which are mutually independent (Fig. 1a). Hydroquinone, as a dual-functional group with both REDOX- and pH-responsive activity, perfectly fits with TPE, a traditional AIEgen, in structure and could be used to introduce into TPE, but has not been studied yet.

Here we report for the first time an intrinsic AIE-active core 1-hydroquinol-1,2,2-triphenylethene (HQTPE) bearing hydroquinone moiety that responds to both oxidants and specific pH value (Fig. 1b, c). HQTPE inherits the AIE properties of the traditional TPE motif. The hydroquinone–quinone transformation and protonation–deprotonation of phenolic hydroxyl endow HQTPE with redox and pH responsiveness, respectively. HQTPE was prepared via a simple palladium-catalyzed cross-coupling between hygroquinone and triphenylethene moieties. Such an intrinsic incorporation of hydroquinone renders HQTPE with an ultrasensitive response mode, which has also been established through theoretical calculation. We also demonstrate that HQTPE is effective in detecting even trace amount of oxidants and has the potential to become an antioxidant or stabilizer in industrial and biomedical applications.

## Results and discussion
### Synthesis and characterization.
As shown in Fig. 2, HQTPE was synthesized for the first time by a two-step reaction with a satisfactory yield from the precursor 1-(2,5-dimethoxyphenyl)-1,2,2-triphenylethene (DMBTPE), whereas its oxidized derivative 1-benzoquinol-1,2,2-triphenylethene (QTPE) was prepared using sodium periodate as an oxidant. All these compounds were characterized by nuclear magnetic resonance (NMR) and mass spectrometry (MS) (please see Supplementary Information Figs. S1–S8). Fourier-transform infrared (FT-IR) spectra provided strong evidences for some key structures such as phenolic hydroxyl groups in HQTPE or carbonyl groups in QTPE (Supplementary Fig. S8). The proposed structures are also in agreement with elemental analysis.

The optical properties of HQTPE and QTPE were studied by ultraviolet-visible (UV-vis) absorption and fluorescence spectroscopy. HQTPE in tetrahydrofuran (THF) solution exhibits only one band located at around 300 nm in the absorption spectra (Supplementary Fig. S10). QTPE shows similar absorption at 300 nm with another new band situated at around 445 nm, resulting in the ochre appearance of QTPE, both in solution or aggregate state (Supplementary Fig. S13).

Fluorescence intensities of HQTPE solutions with different water (poor solvent)/THF (good solvent) ratios ($f_w$) were utilized to investigate the AIE characteristics. As shown in Fig. 3, HQTPE behaves as a faint emitter in THF solution and there is no obvious change in fluorescence intensity until $f_w$ reaches 90%. When $f_w$ is >90%, the intensity increases significantly with further addition of water, suggesting that HQTPE is typical of the AIE-active luminogen. Scanning electron micrographs provided a closer observation that HQTPE was as expected in a state of aggregation when fluorescing (Supplementary Fig. S12); most are micrometer-sized particles precipitated from the 99 vol% water/THF solvent. Then we tested the fluorescence of HQTPE dispersed in different solvents (Supplementary Fig. S11). The results also pointed to its AIE performance. In the solvent with poor solubility (e.g., water) or high viscosity (e.g., PEG-200), HQTPE had a significant increase in fluorescence intensity.

### Oxidation-induced intermolecular quenching effect.
HQTPE is a unique redox-active AIEgen-bearing hydroquinone moiety. The hydroquinone–quinone transformation endows HQTPE with switchable oxidation states. A series of solutions with gradient ratios of HQTPE/QTPE was prepared to simulate the oxidation process of HQTPE. Figure 4a shows the fluorescence spectra of each system, where the fluorescence decays as the oxidation progresses. We obtained Fig. 4b by fitting their fluorescence intensities at 465 nm with the corresponding degrees of oxidation. There is a nonlinear relationship between fluorescence and the degree of oxidation. As the oxidation product QTPE contains a fluorescence-quenching unit, $p$-benzoquinone[23], even only trace amount of HQTPE is oxidized into QTPE, the fluorescence should be sharply attenuated. When the oxidation degree reached 0.20, almost no fluorescence could be detected, whereas the color of the solution shows little change (as shown in Fig. 4c). During an ongoing oxidation process, the attenuation of fluorescence signal will not be proportional to the consumption of HQTPE, but much faster than the oxidation process. Newly generated QTPE has an amplification effect on fluorescence attenuation by its quenching effect, which accounts for the extremely high sensitivity of HQTPE. With this fascinating effect, HQTPE could be further exploited as an innovative oxidation indicator. We will elaborate in the application section how this intermolecular quenching effect works.

To acquire comprehensive understanding of the fluorescence features, related theoretical calculations were performed (Fig. 4d). Density functional thoery (DFT) calculations in the solution phase uncover that significant decreasing of oscillator strength

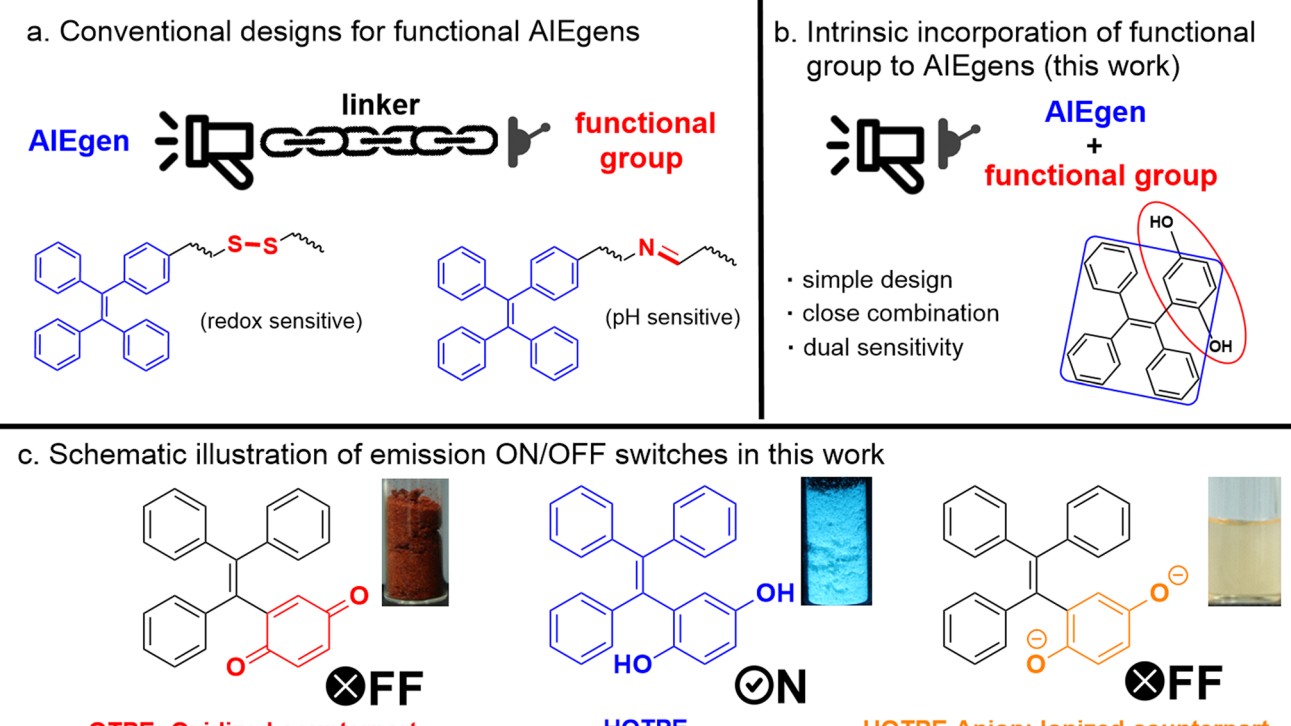

**Fig. 1 Conventional designs and intrinsic functionalization for AIEgens. a** Conventional designs for functional AIEgens. **b** Intrinsic functional AIE-active core in this work. The functional group hydroquinone and AIEgen TPE share part of the structure. **c** Schematic illustration of emission ON/OFF switches.

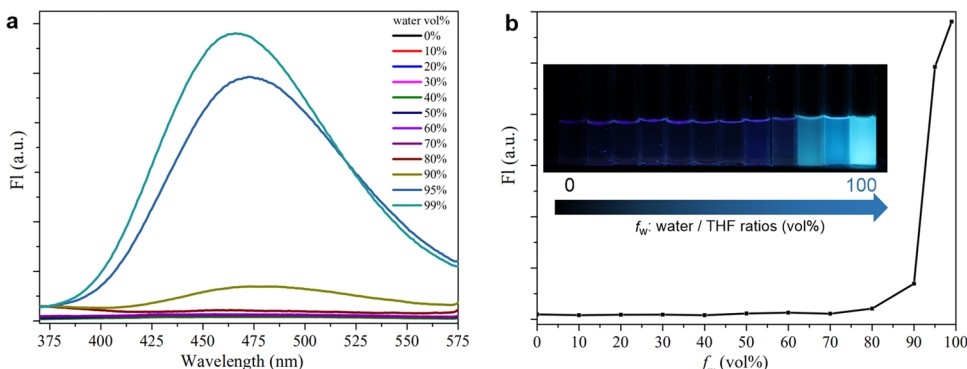

**Fig. 2 Synthesis of 1-(2,5-dimethoxyphenyl)-1,2,2-triphenylethene (DMBTPE) and 1-hydroquinol-1,2,2-triphenylethene (HQTPE).** HQTPE could be synthesized by Suzuki–Miyaura coupling and demethylation with a satisfactory yield.

**Fig. 3 AIE properties of HQTPE. a** Fluorescence spectra of 100 μM HQTPE in THF/Water mixtures with different water fractions excited at 300 nm. **b** Plot of the fluorescence intensity of HQTPE at different THF/Water ratios. Inset shows the photograph of solutions of HQTPE at various THF/Water compositions upon exposing to UV light.

between $S_1$ and $S_0$ in oxidation species QTPE. Classical theory associates radiation rate of fluorescence with its molecular oscillator strength[24]. Further, decreasing oscillator strength slows down the radiation rate of QTPE, which explains for the reduced luminescence efficiency. Then, through the natural transition orbitals analysis by multiwfn[25], we found the weaker oscillator strength of QTPE compared to HQTPE mainly originates from the absence of the overlap between the donor and acceptor orbitals at the center double bond area (for detailed analysis, see Supplementary Figs. S21 and S22, and Supplementary Table S1).

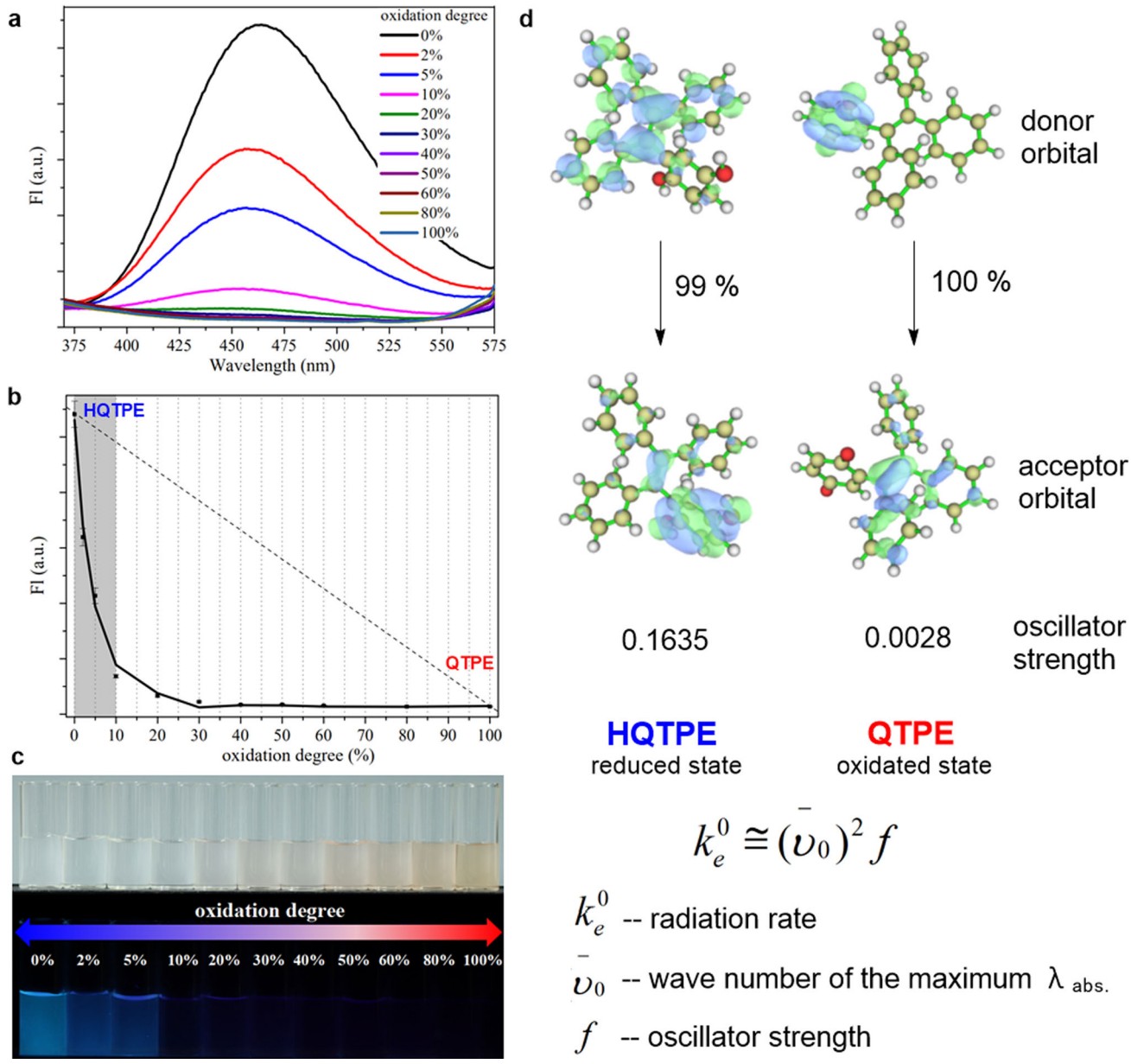

**Fig. 4 Intermolecular fluorescence quenching of HQTPE during oxidation, simulated by separately adding HQTPE and QTPE at various ratios.**
**a** Fluorescence spectra of 100 µM HQTPE/QTPE with different degrees of oxidation, at excitation wavelength of 300 nm. **b** Fitting curve of the fluorescence intensity of HQTPE/QTPE solutions with different degrees of oxidation. Error bars show SDs of three independent measurements. **c** Photographs of HQTPE/QTPE solutions with different degrees of oxidation under white light or UV light. **d** The natural transition orbitals (NTOs) pairs describing $S_0$–$S_1$ transition and corresponding oscillator strength of HQTPE and QTPE. Numbers near the arrow indicate the contribution percentage of displayed NTO transitions to the corresponding $S_0$–$S_1$ transitions. Calculated at optimized $S_1$ geometry using TD-DFT method.

Moreover, we have also calculated the adiabatic excitation energy of $S_0$–$S_1$ transition in HQTPE and QTPE. We find QTPE ($E_{ad} = 1.34$ eV) have significantly smaller adiabatic excitation energy than HQTPE ($E_{ad} = 2.98$ eV) (some other photochemical properties of related molecules are shown in Supplementary Table S3). According to the energy gap law, smaller energy gap will result in faster nonradiative decay process between the involving states. The faster nonradiative decay process from $S_1$ to $S_0$ in QTPE compared with HQTPE will result in shorter lifetime and lower quantum yield, which also contribute to the oxidation-induced intermolecular quenching effect of HQTPE.

The fluorescence-quenching efficiency was calculated using the Stern–Volmer equation $I_0/I = 1 + K_{sv}[\text{QTPE}]$ to quantitatively describe the quenching behavior in the presence of QTPE (where

$I_0$ is the initial fluorescence intensity of HQTPE in $H_2O$, $I$ is the fluorescence intensity in the presence of QTPE, and $K_{sv}$ is the Stern–Volmer quenching constant). The plot of $I_0/I$ against the concentration of QTPE (Fig. 5a) showed a drastic jump at the initial stage (from 0 to 15 mol%), rapidly followed by the maximum quenching, which was consistent with the above phenomenon. The Stern–Volmer constant $K_{sv}$ at the initial stage was estimated to be $1.524 \times 10^5$ M$^{-1}$ for QTPE. The $S$–$V$ plot suggested that the fluorescence change through oxidation involves a self-quenching process and the obtained $K_{sv}$ value described this process in more detail.

The quenching behavior was also examined with quantum yield and lifetime measurements. The quantum yield value of HQTPE in the absence of QTPE in $H_2O$ was found to be 0.104, but

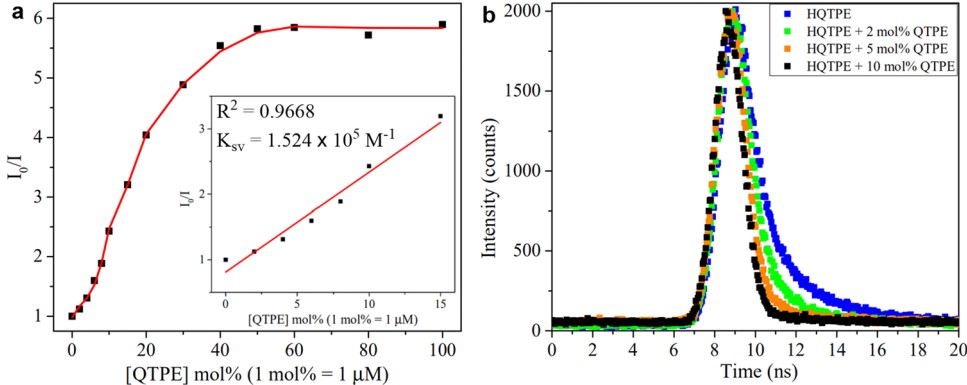

**Fig. 5 Quantitative discussion on fluorescence quenching of HQTPE in the presence of QTPE. a** $K_{sv}$ plot for 100 μM HQTPE in THF-H$_2$O (vol 1% : 99%) upon separate addition of QTPE (0–100 mol%), at an excitation wavelength of 300 nm. Inset: a piecewise linear fitting curve at the initial stage (from 0 to 15 mol% QTPE). **b** Fluorescence lifetime decay plots of HQTPE in the absence and presence of QTPE.

**Table 1 Quantum yield and lifetime parameters of HQTPE in THF-H$_2$O (vol 1% : 99%) in the absence and presence of QTPE.**

| Sample | φ | $\tau_{avg}$ (ns) | $K_r$ (ns$^{-1}$) | $K_{nr}$ (ns$^{-1}$) |
|---|---|---|---|---|
| HQTPE | 0.104 (in water) | 1.87 | 0.056 | 0.479 |
| | 0.213 (in solid state) | | | |
| HQTPE + 2 mol% QTPE | 0.040 | 0.693 | 0.057 | 1.385 |
| HQTPE + 5 mol% QTPE | 0.025 | 0.555 | 0.045 | 1.757 |
| HQTPE + 10 mol% QTPE | 0.008 | 0.321 | 0.025 | 3.090 |

decreasing to 0.040, 0.025, and 0.008, respectively, in the presence of QTPE with different concentrations (summarized in Table 1). The drastic drop in the quantum yield values ensured the fluorescence quenching due to the interactions between HQTPE and its oxidized counterpart. Further, the fluorescence decay behavior was assessed using time-resolved spectrofluorometry and the results are given in Fig. 5b. The average fluorescence lifetime of HQTPE in H$_2$O was found to be 1.87 ns, which became shorter in the presence of QTPE. We also estimated the radiative ($k_r$) and nonradiative ($k_{nr}$) decay values summarized in Table 1. The calculated $k_{nr}$ showed a significant grow, whereas $k_r$ dropped down with a small quantity of QTPE. The reasonable reduction of radiative process and enhancement of nonradiative process resulted in a sharp fluorescence-quenching behavior during the oxidation of HQTPE.

**Response to the pH of media**. It is well known that the aromatic hydroxyl group behaves as a weak acid in aqueous media (typical p$K_a$ value of aromatic hydroxyl is about 10). Therefore, we attempt to explore the pH-regulated fluorescence performance of HQTPE with two para-substituted phenolic hydroxyl group. The fluorescence changes of HQTPE at different pH values from 0.56 to 13.24 are shown in Fig. 6a and the fitting curve is plotted in Fig. 6b. Despite the existence of excited-state proton transfer and deaggregation due to protonation in acidic conditions, HQTPE still exhibits bright emission and the fluorescence intensity increases until the pH value reaches about 10.00, when the deprotonation starts. In basic media with pH value of >10, the fluorescence intensity dramatically decreases to near zero. This transition process is also accompanied with visible color change as shown in Fig. 6c.

In addition, theoretical calculations showed the differences of overlap and a decreasing trend of oscillator strength with HQTPE and the corresponding anionic species (Fig. 6d), which could not fluoresce in an anionic state. Detailed analysis is given in Supplementary Fig. S23 and Supplementary Table S2. Furthermore, similar to the oxidation species QTPE, we found the

corresponding anionic species of HQTPE also have smaller adiabatic excitation energy than HQTPE ($E_{ad} = 1.38$ eV for mono-deprotonated species, $E_{ad} = 0.66$ eV for di-deprotonated species, and $E_{ad} = 2.98$ eV for HQTPE). This will lead to faster nonradiative decay process according to the energy gap law, which could be another reason contributing to the pH-response behavior of this system.

In contrast, the precursor DMBTPE possessing only two proton receptors but no active protons was insensitive to pH changes. Fluorescence intensity of DMBTPE shows less variation in the pH range of 1–13 as compared with HQTPE (Supplementary Fig. S15), which validates the essential role of HQ moiety in a pH-responsive process. These results indicate the superb pH sensitivity of HQTPE and its potential use in alkalinity monitoring.

**Electrochemical measurement**. The electrochemical studies of HQTPE could further evaluate its redox behavior and electroactivity, and reveal potential applications in electronic devices. We investigated the electrochemical performance of HQTPE in comparison with other reference substances by cyclic voltammetry scanning in propylene carbonate (PC) solution containing tetrabutyl ammonium hexafluorophosphate ($^tBu_4NPF_6$) as the supporting electrolyte. All the measurements were made at a solution concentration of $10^{-3}$ M at room temperature. Half-wave potential values [$E_{1/2} = (E_{pa} + E_{pc})/2$] were calibrated by that of the internal reference Fc/Fc$^+$ couple (standard $E_{1/2} = 0.492$ V, found 0.493 V).

Figure 7a shows the CV curves of HQTPE and other reference substances at a scanning rate of 100 mV s$^{-1}$. Through comparison to the CV curves of pure electrolyte (PC, ferrocene, and hydroquinol-benzoquinone as controls), HQTPE exhibits a symmetric redox couple with oxidation peak at −0.061 V and reduction peak at −0.695 V vs. Ag/AgCl. To further investigate the effect of triphenylethene moiety, we carried out CV scanning of $p$-hydroquinone under the same condition. The oxidation and reduction peak of HQ located at 0.070 V and −0.311 V, respectively with $E_{1/2} = -0.121$ V, which is higher than that of

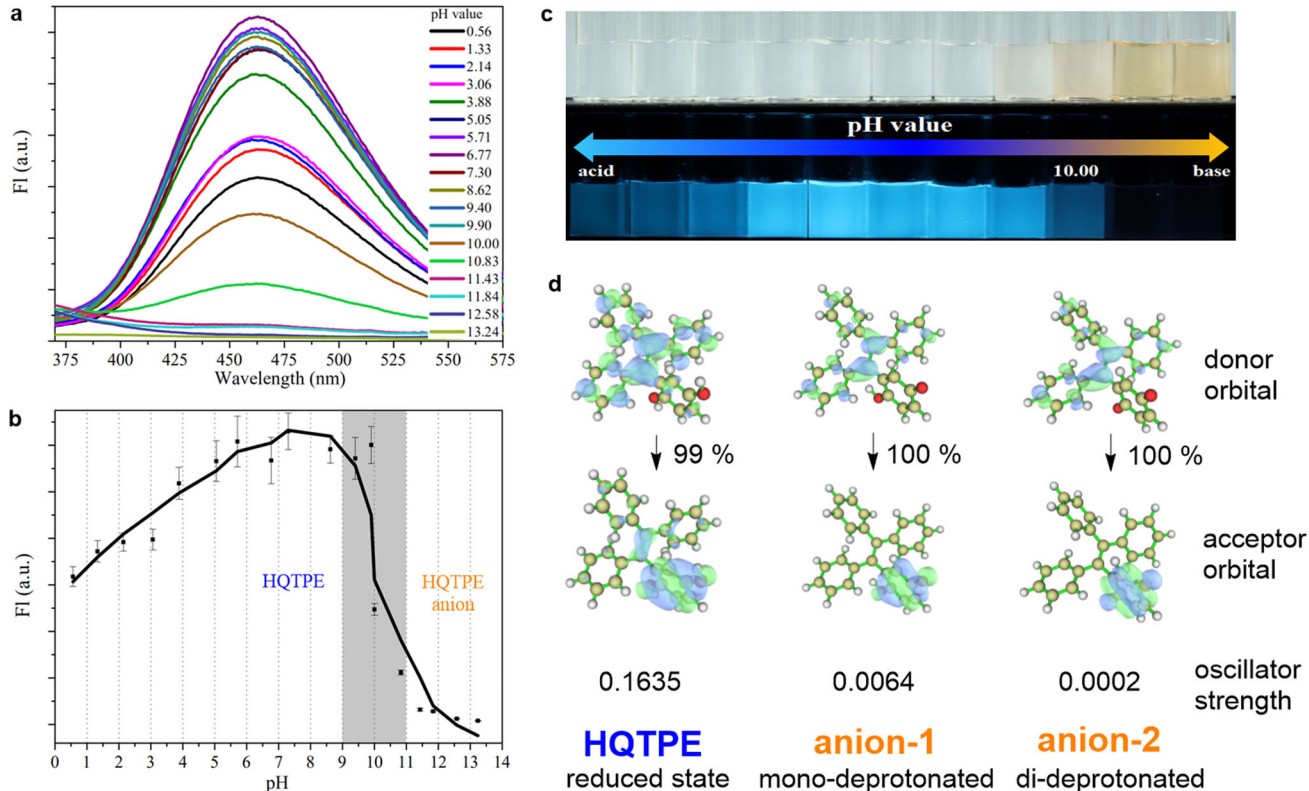

**Fig. 6 Detection of pH value. a** Fluorescence responses of 100 μM HQTPE to different pH values at excitation wavelength of 300 nm. **b** Fitting curve of the fluorescence intensity of HQTPE solutions under various pH conditions. Error bars show SDs of three independent measurements. **c** Photographs of HQTPE solutions at different pH conditions under white light or excited by UV light. **d** The natural transition orbitals (NTO) pairs describing $S_0$–$S_1$ transition and corresponding oscillator strength of HQTPE and corresponding anions. Numbers near the arrow indicate the contribution percentage of displayed NTO transitions to the corresponding $S_0$–$S_1$ transitions. Calculated at optimized $S_1$ geometry using TD-DFT method.

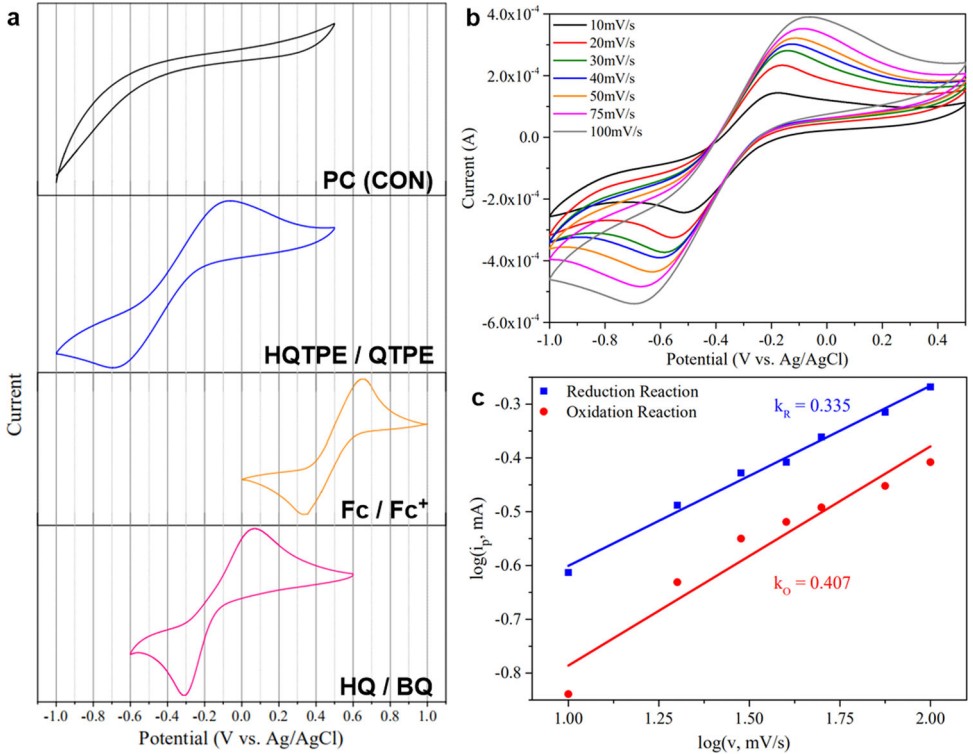

**Fig. 7 Electrochemical performance of HQTPE. a** CV curves of HQTPE and other reference substances: pure PC, unsubstituted ferrocene, and *p*-hydroquinone at a scanning rate of 100 mV s$^{-1}$. **b** CV curves of HQTPE at different scan rates. **c** Corresponding log $i_p$ vs. log $v$ plot of HQTPE.

HQTPE. It shows the fact that the strong electron-donating effect of the triphenylethene substituent in HQTPE enables faster electron transfer. And as a result, HQTPE possesses stronger reducibility and higher responsibility to oxidants. Moreover, the peak separation of HQTPE is wider than hydroquinone, revealing poorer electrochemical reversibility due to the strong tendency towards an oxidized form. With this molecular design, the sensitivity and redox activity is notably promoted with modest sacrifice of chemical reversibility.

Figure 7b shows the kinetic behavior of HQTPE with CV measurement at varying scan rates. The voltammetric responses of HQTPE at different scan rates can be readily described with equation $\log i_p = k \log v^{26}$, where $i_p$ is the peak current value and $v$ is the scan rate. As shown in Fig. 7c, an excellent linear relationship between $\log i_p$ and $\log v$ is obtained. Slopes of the plots for HQTPE are 0.335 and 0.407 for reduction and oxidation processes, respectively, which is indicative of the characteristic quadratic of diffusion control. The larger slope value in the oxidation process indicates faster kinetics than the reduction process, which confirms again the remarkable redox activity of HQTPE.

**Application**. Based on its high reactivity and sensitivity, we have explored some potential applications of HQTPE in detecting diverse oxidants at various concentrations. Oxidant level has been one of the most common detection index because of its significance in the fields of environment, food security, and health care. We first tested HQTPE with its universal capability for the detection of various oxidants. As shown in Fig. 8a, the initial reduction state of HQTPE exhibits high fluorescence intensity. With the addition of oxidants, subsequent oxidation of HQTPE occurred and the quenching agent QTPE is generated, resulting in rapid fluorescence intensity decrease. HQTPE responds to both the two major classes of oxidants, hyperoxides (TBHP and $H_2O_2$ for instance) and oxidative ions ($MnO_4^-$, $IO_4^-$, and $CrO_4^{2-}$ for instance), in a well-behaved manner. For ROS such as singlet oxygen ($^1O_2$), HQTPE could also show great detection and clearance effect in situ, which was reflected by electron paramagnetic resonance (EPR) signals (Supplementary Fig. S16). Then we examined the limit of detection by adding different concentrations of oxidant ($H_2O_2$ for instance). We even found 0.25 μM hydrogen peroxide (2.5 mol% relative to HQTPE) still triggered response because of the intermolecular quenching effect arising from the quinoid product QTPE. This establishes HQTPE with having responsiveness to even a trace amount of oxidant that existed. In Supplementary Fig. S17, we further explored the detection limits for different oxides. For both ionic and peroxygen-type oxidants, fluorescence signal changes were realized at a lower material ratio and concentration. Meanwhile, the species with higher oxidation value showed a more obvious fluorescence change, which is in line with the trend of the stoichiometry of the REDOX reaction. These results demonstrated the potential of HQTPE for rapid and ultrasensitive oxidant detection.

In addition to serving as a redox probe, the oxygen scavenging and detecting capability of HQTPE has also been studied. As oxygen is one of the most important factors causing spoilage and corrode in food, medicine and materials, and antioxidant has attracted extensive research interest. To demonstrate the high reactivity of HQTPE to oxygen at ambient condition, we have selected tert-butylhydroquinone (TBHQ) as a reference compound, which is one of the best commercial antioxidants and stabilizers sharing similar structure with HQTPE. As shown in Fig. 9, HQTPE exhibits more efficient deoxygenation ability than TBHQ both in solid (Fig. 9a) and in solution (Fig. 9b) states under ambient condition, which is evidenced quantitatively by $^1H$ NMR spectra (Fig. 9c). Focusing on the phenolic hydroxyl signal, about 43% of solid HQTPE have reacted with oxygen within 7 days at room temperature, whereas TBHQ has hardly been oxidized (Supplementary Figs. S18 and S19). Besides, the deoxidizing of HQTPE is accompanied with both visible color change and fluorescence quenching, suggesting a unique advantage over the commonly used iron and enzyme series of deoxidizers. The high deoxygenation activity and novel response mode renders HQTPE with great potential in future chemical and biological (in vivo or in vitro) antioxidant applications.

## Conclusion

We have described a simple but novel intrinsic AIE-active core HQTPE with dual responsiveness through the intrinsic incorporation of hydroquinone moiety to TPE fluorophore for the first time. Such an intrinsic modification of TPE molecular structure brought about highly sensitive response to common oxidants and specific pH values, which can even be distinguished with the naked eyes. The redox activity is validated via electrochemistry and the redox-induced photo-physical behavior is elucidated via theoretical calculations. With the analysis of electronic state and oscillator strength, we have established a rational mechanism to explain the resultant luminescent behaviors of hydroquinone-functionalized AIEgens. Moreover, we have applied HQTPE for sensitive detection and efficient scavenging of oxidants. This work should offer a new strategy for designing intrinsic and electro-active AIE cores with multi-responsiveness and also broaden the scope of conventional AIEgens. It is noteworthy that other

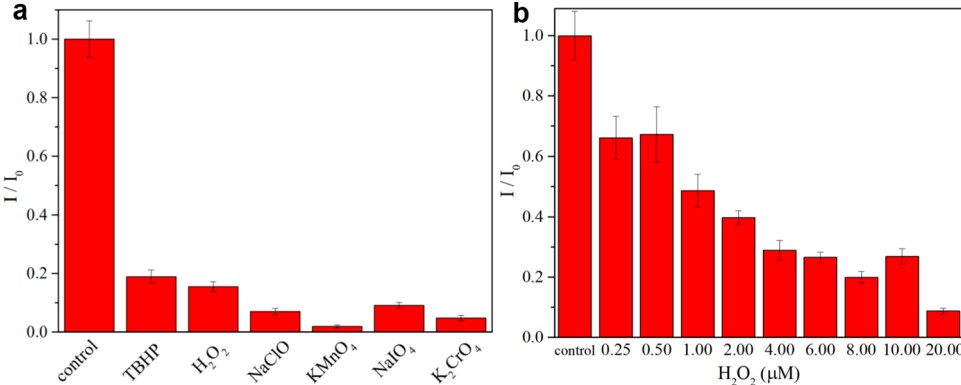

**Fig. 8 Efficient response of HQTPE to oxidants. a** Relative fluorescence intensity at 462 nm of 10 μM HQTPE in water before and after the addition of 10 μM oxidants or **b** $H_2O_2$ at various concentrations. $I_0$ is the fluorescence intensity of 10 μM HQTPE in water at 462 nm (control) and $I$ is the corresponding fluorescence in the presence of testing species with gradient concentrations. Error bars show SDs of three independent measurements.

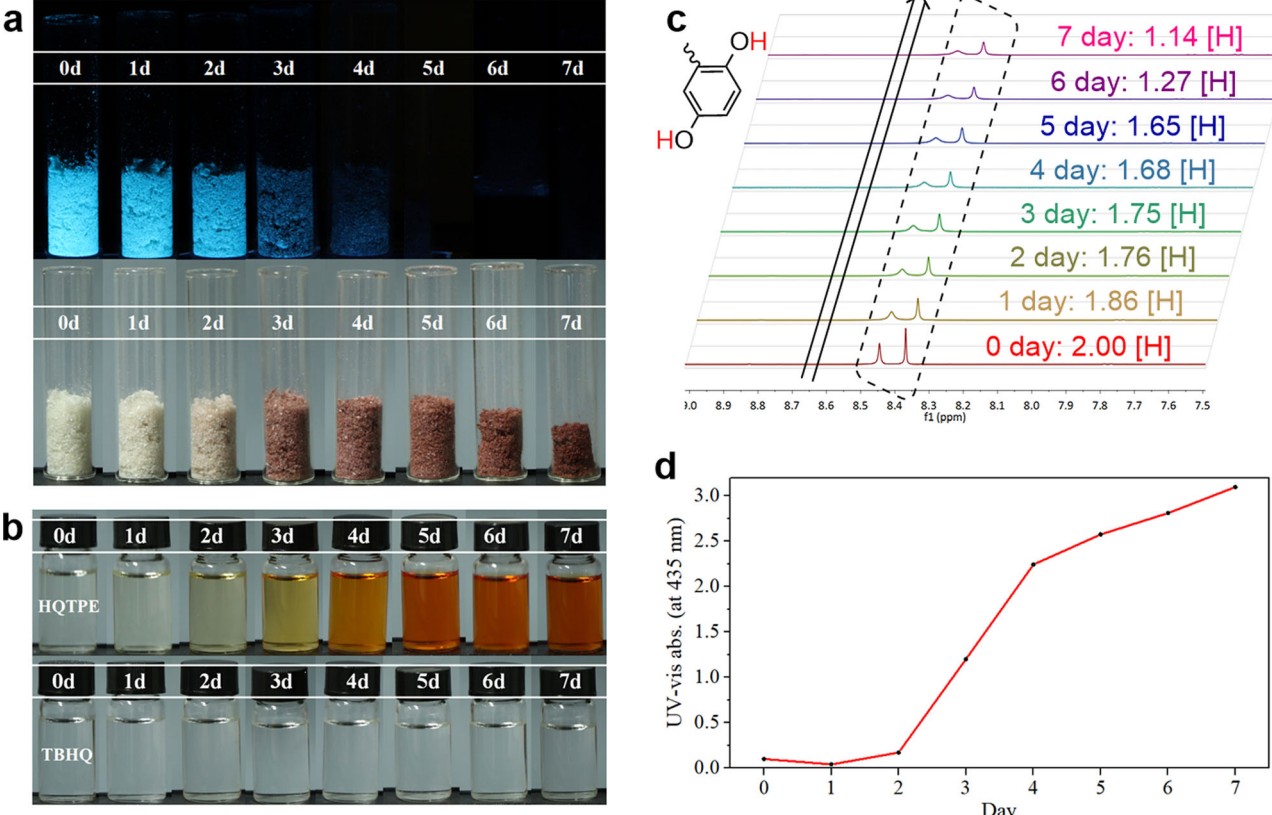

**Fig. 9 Oxidation process of HQTPE with exposure to ambient condition. a** Photographs of HQTPE powder exposed to air within 7 days under exposure to 300 nm UV light or white light. **b** Photographs of HQTPE and TBHQ solutions exposed to air within 7 days under white light. HQTPE exhibits much better deoxygenation activity than TBHQ. **c** $^1$H NMR spectras in DMSO-d$_6$ of HQTPE solution. There was a clear decrease of phenol hydrogens at 8.4 p.p.m. after the week-long exposure to air. The same test was also performed on TBHQ, thus ruling out the possibility of hydrogen-deuterium exchange. **d** Change in the intensity of absorption band at 435 nm of HQTPE solution upon exposure to air within 7 days.

functional groups including polymerizable vinyl moieties could be incorporated in HQTPE for a variety of chemical and biological applications. Further investigation is in active progress and the results will be reported in due time.

## Methods

**Materials and procedures**. The carbon–carbon coupling, demethylation, and diphenol oxidation reactions were all performed following previously published procedures (Fig. 2)[27–29]. All the commercially available chemicals and solvents were used directly without further purification, unless specified otherwise. Reactions were monitored using thin layer chromatography (TLC) with commercial TLC plates. Products were isolated by silica gel column chromatography.

**Instrumentation**. All $^1$H and $^{13}$C NMR spectra were recorded in DMSO-d$_6$ solutions at 298 K using a Bruker (400 MHz) spectrometer with tetramethylsilane as an internal standard. Mass spectra were recorded on matrix-assisted laser desorption ionization time-of-flight MS (MALDI-TOF MS), Performance (Shimadzu, Japan). Perkin Elmer CE-440 was used for micro elemental analyses. Infrared spectra were recorded by Perkin Elmer (Spectrum 100) FT-IR Spectrometer. UV-vis spectra were recorded using Perkin Elmer (Lambda 750) UV/Vis/NIR Spectrometer in ethyl acetate solvent. Fluorescence emission, quantum yield, and lifetime measurements were performed by SHIMADZU Spectro fluorophotometer RF-6000. Aggregations of HQTPE were examined by scanning electron microscopy (SEM) by a Hitachi SU-8010 SEM instrument fitted with a field-emission source and operating at 10 kV. Samples were prepared for SEM by mounting a portion of the cover slide on a silicon stub. Electrochemical measurements were performed on a CH-Instruments Model CHI660D using a one-compartment cell under air atmosphere. The three-electrode system consisted of a platinum counter electrode, an Ag/AgCl reference electrode, and a platinum working electrode.

**Synthesis of DMBTPE, HQTPE, and QTPE**. DMBTPE was synthesized by Suzuki–Miyaura coupling with a final yield of 89.2% (Supplementary Note 1 and Scheme S1). M.p. 155–157 °C. $^1$H NMR (400 MHz, DMSO-d$_6$): δ7.12–7.01 (m, 9H, of three benzene rings), 6.93 (ddd, J = 9.4, 7.6, 2.0 Hz, 6H, of three benzene rings),

6.73 (d, J = 8.9 Hz, 1H, of the DMB ring), 6.65 (dd, J = 8.8, 3.1 Hz, 1H, of the DMB ring), 6.51 (d, J = 3.1 Hz, 1H, of the DMB ring), 3.50 (s, 3H, of methoxyl), 3.36 (s, 3H, of methoxyl). $^{13}$C NMR (101 MHz, DMSO-d$_6$): δ153.17, 151.83, 143.89, 143.10, 142.81, 141.61, 137.69, 133.83, 131.18, 130.35, 130.08, 128.33, 128.03, 127.98, 127.06, 127.01, 126.69, 117.90, 113.30, 56.40, 55.72. MALDI-TOF MS: calcd for C$_{28}$H$_{24}$O$_2$ [M]$^+$, 392.18; found, 392.32.

HQTPE was further synthesized by demethylation of DMBTPE with a final yield of 83.5% (Supplementary Note 1 and Scheme S1). M.p. 198–200 °C. FT-IR: ν (OH) 3524 (m) and 3409 (m), ν(arC-H) 1176 (s), 770 (s), 748 (s) and 697 (vs), ν (arC-C) 1489 (s), and 1443 (s) cm$^{-1}$. $^1$H NMR (400 MHz, DMSO-d$_6$): δ8.44 (s, 1H, of phenolic hydroxyl), 8.37 (s, 1H, of phenolic hydroxyl), 7.15–6.96 (m, 12H, of three benzene rings), 6.94–6.91 (m, 3H, of three benzene rings), 6.43 (d, J = 8.2 Hz, 1H, of the hydroquinone ring), 6.36–6.30 (m, 2H, of the hydroquinone ring). $^{13}$C NMR (101 MHz, DMSO-d$_6$): δ149.67, 148.22, 144.08, 143.49, 143.11, 141.01, 138.54, 131.49, 131.30, 130.47, 130.23, 127.99, 127.82, 126.89, 126.44, 118.20, 116.41, 115.19. MALDI-TOF MS: calcd for C$_{26}$H$_{20}$O$_2$ [M]$^+$, 364.15; found, 364.17. Anal. calcd for for C$_{26}$H$_{20}$O$_2$: C, 85.69; H, 5.53; O, 8.78%. Found: C, 84.80; H, 5.64; O, 8.21%.

QTPE is the ultimate oxidation product of HQTPE. Sodium periodate (NaIO$_4$), as the most effective one of various available oxidants, was chosen for large-scale preparation of QTPE, with a final yield of 98.9% (Supplementary Note 1 and Scheme S2). M.p. 167–168 °C. FT-IR: ν(C = O) 1651 (m), ν(arC-H) 1080 (s), 768 (s), 736 (s) and 697 (vs), ν(arC-C) 1444 (s) and 1281 (s) cm$^{-1}$. $^1$H NMR (400 MHz, DMSO-d$_6$): δ7.27–7.18 (m, 3H, of three benzene rings), 7.14–7.12 (m, 3H, of three benzene rings), 7.10–7.04 (m, 3H, of three benzene rings), 7.04–7.01 (m, 3H, of three benzene rings), 6.94 (tt, J = 5.8, 3.4 Hz, 3H, of three benzene rings), 6.76–6.67 (m, 2H, of the benzoquinone ring), 6.49 (d, J = 2.3 Hz, 1H, of the benzoquinone ring). MALDI-TOF MS: calcd for C$_{26}$H$_{18}$O$_2$ [M + H]$^+$, 363.14; found, 363.21. Anal. calcd for for C$_{26}$H$_{18}$O$_2$: C, 86.16; H, 5.01; O, 8.83%. Found: C, 85.10; H, 5.28; O, 8.46%.

**Computational methods**. Unless otherwise specified, all DFT calculations were performed with Gaussian 16[30]. The geometry optimizations were performed in the gas phase with the B3LYP-D3 functional and 6–31 G(d) basis for neutrals and cations, and in the solution (THF, conductor-like polarizable continuum model (cpcm) solvation model) phase with the B3LYP-D3 functional and 6–31 + G(d)

basis for anions. Energies and oscillator strengths were calculated with the B3LYP-D3 functional and 6–311 + G(d,p) basis, using the cpcm solvation model with THF as the solvent. Excited states were calculated through TD-DFT method. (Each $S_1$ was optimized and oscillator strength showed in the main study are all obtained from optimized $S_1$ geometry.) See more computational details in Supplementary Note 2 and Supplementary Data 1–8.

## Data availability
The data supporting the finding of this study are available in the article and Supplementary Information or from the lead contact upon reasonable request. Further information and requests for resources and reagents should be directed to and will be fulfilled by the lead contact, Yen Wei (weiyen@tsinghua.edu.cn). DMBTPE and HQTPE generated in this research will be made available on request, but we may need a payment and/or a completed Materials Transfer Agreement if there is potential for commercial application.

## Code availability
The codes supporting the computational studies of this study are available in the Supplementary Information or from the lead contact upon reasonable request.

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

## Acknowledgements
This work was supported by the National Natural Science Foundation of China (numbers 21788102 and 24174057). We are most grateful to Danning Hu, Liucheng Mao, Junyu Chen, Hongye Huang, and Professor Lei Tao of Tsinghua University for valuable assistance, advice, and suggestions.

## Author contributions
Y.W., Z.-G.S., and J.-Y.Y. conceived and supervised the project. M.-S.W. and R.-J.H. carried out the synthesis, characterization, and other experimental work. Y.-H.W. performed the theoretical calculation and analysis. M.T. and X.-Y.Z. provides convenience and guidance for the use of instruments. M.-S.W., Y.-H.W., and R.-J.H. wrote the paper with input from all authors.

## Competing interests
The authors declare no competing interests.
