## [Peer Review File · Communications Chemistry]

Reviewers' comments:

Reviewer #1 (Remarks to the Author):

In this manuscript, the authors prepared 1-Hydroquinol-1,2,2-triphenylethene compound as an AIE core with dual sensitivity. This compound is more emissive in the solid state than in solution, which exhibits AIE characteristics. This work provides an intrinsic hydroquinone functionalised AIE core with dual sensitivity. In general, I recommend its publication in Communications Chemistry. Here are some comments:

- The melting point should be placed for the synthesised compounds. See carefully the proton constants in NMR (Synthesis of DMBTPE, HQTPE and QTPE).
- Studies of Absorption and Fluorescence in different solvents could be done, because it would help to understand the behaviour of this compound in other solvents and not only in THF and water.
- The measure of the HQTPE fluorescence quantum yield in the solid state should be determined.
- The same for the calculation of the detection and quantification limits of QTPE, for each of the oxidants tested.
- Some important previous works on AIE material can be introduced in the article. For example: Chem. Rev. 2015, 21, 11718-11940; Mater. Chem. Front, 2019.3, 1335-1340, Adv. Optical Mater. 2020 (<https://doi.org/10.1002/adom.202000162>).

Reviewer #2 (Remarks to the Author):

In this paper, authors reported a fluorescent molecule 1-hydroquinol-1,2,2-triphenylethene (HQTPE), which can be regarded as a combination of hydroquinone and tetraphenylethene. This molecule exhibited aggregation-induced emission (AIE) property and its emission was also sensitive to redox and pH value. The working mechanism was studied by control experiments and theoretical calculations. And the potential application of this molecule as a redox probe was proposed. In my opinion, the redox- and pH-sensitive emission properties of HQTPE can be attributed to the presence of hydroquinone, a well-known molecule that is highly prone to oxidation and deprotonation. The introduction of hydroquinone to other dye molecules can also lead to similar redox- and pH-sensitive emission properties. The study on the sensing application is quite superficial. Authors claimed that the AIE property of TPE could solve ACQ problem and give high PLQYs in aggregated state. But the aggregate of HQTPE only gave a low PLQY of ~0.1. On the other hand, HQTPE can be easily oxidized and I do not think HQTPE is of high purity. Actually, the elemental analysis data of HQTPE are poor as shown in SI. I think this work is actually routine and lacks novelty and significance, and there are no advantages of this molecule for pH and oxidant sensing.

Reviewer #3 (Remarks to the Author):

In this manuscript, a new luminescence compound, 1-hydroquinol-1,2,2-triphenylethene(HQTPE) are synthesized and characterized. And its intrinsic redox and pH activities were investigated and tested. The manuscript stated some interesting experimental phenomena that there are great differences of photochemical properties between HQTPE, the oxidized counterpart QTPE and its anions, and explained that HQTPE shows rapid quenching of fluorescence as the increase of the solution concentration. The dependence of the fluorescence of HQTPE and reducibility, and pH value of media were studied. But in the present manuscript, there are still several issues to be clarified:

1. In Page 6 line 108-110 the authors mentioned "During the oxidation process, newly generated QTPE has an amplification effect on fluorescence quenching, which accounts for the extremely high sensitivity of HQTPE." How to quantify the amplification on fluorescence quenching effect of newly generated QTPE?

2. The Figures 2A, 2D, 4A and 4D seem to be blurry.

3. There are lack of detailed information of $S_1 \rightarrow S_0$ transition and corresponding configuration interaction coefficients.

Reviewer #1:

#The melting point should be placed for the synthesised compounds. See carefully the proton constants in NMR (Synthesis of DMBTPE, HQTPE and QTPE).

ANSWER: The melting point data for each substance have been updated in the synthesis section. At the same time, we also checked all the NMR data again, and they are consistent with their respective chemical structures. Thank you.

#Studies of Absorption and Fluorescence in different solvents could be done, because it would help to understand the behaviour of this compound in other solvents and not only in THF and water.

ANSWER: In the SI section (Fig. S11), we added the fluorescence intensities of **HQTPE** in ten common solvents (water, chloroform, methanol, ethanol, tetrahydrofuran, N-methylpyrrolidone, PEG-200, petroleum ether, n-hexane, etc.), and make a comparison in the same figure. The fluorescence intensity also reflects the solubility of the AIE core in these different solvents, which is also consistent with its inherent AIE property. Thank you.

#The measure of the HQTPE fluorescence quantum yield in the solid state should be determined.

ANSWER: This data has been updated. PLQY of solid **HQTPE** is about 0.213. See Table 1 on page 8 for details. By measuring the fluorescence quantum yield in the solid state, the problem of incomplete aggregation in the bad solvent was remedied, and the new data also proved that **HQTPE** has a small amount of dissolved parts in water. Thanks for your advice.

#The same for the calculation of the detection and quantification limits of QTPE, for each of the oxidants tested.

ANSWER: Thank you for advice. Relevant experiments have been supplemented in SI, please refer to page 12 in manuscript and Fig. S15 for details. By measuring the detection limits of different oxidants, we also found some new phenomena. For example, the specie with higher oxidation value shows lower detection limits when using **HQTPE** for trace oxidant detection, which is in line with the trend of the stoichiometry of the REDOX reaction.

#Some important previous works on AIE material can be introduced in the article. For example: Chem. Rev. 2015, 21, 11718-11940; Mater. Chem. Front, 2019.3, 1335-1340, Adv. Optical Mater. 2020 (<https://doi.org/10.1002/adom.202000162>).

ANSWER: We did miss some important works, and two related papers here have been referred, namely Ref. 9 and 17. Thank you.

Reviewer #2 :

// In my option, the redox- and pH-sensitive emission properties of HQTPE can be attributed to the presence of hydroquinone, a well-known molecule that is highly prone to oxidation and deprotonation. The introduction of hydroquinone to other dye molecules can also lead to similar redox- and pH-sensitive emission properties. The study on the sensing application is quite superficial.

ANSWER: The main content of this paper includes the following parts. One is about the intrinsic functionalization concept in molecular design. Functional group could integrate with the fluorophore skeleton by the similarity in structure between hydroquinone and tetraphenylethylene. That is why in the title of manuscript marked "An Intrinsic Hydroquinone-Functionalized AIE Core" and "Integrated Molecular Design". This new idea of integration is completely different from the traditional stitching methods. The second is to explore the properties and applications of such a new AIE core. As we can

see, the application is not the most important but still exciting part of the manuscript. For example, its oxygen affinity is far better than the current commercial deoxidizer, and its detection and removal of reactive oxygen species will play a role in the field of biomedicine. Meanwhile, it also has the property of electrochemical stimulus response. We have also completed some more exciting application around the intrinsic-functionalized AIE core in electrode visualization and Li-electrode probe, but due to the limitation of space and the relevance of content, it was not shown in the same paper but as another independent work. We still think that as the beginning of a series of work, molecular design and the concept of intrinsic functionalization are the top priorities of this paper. We hope that reviewer will pay attention to it. Thank you.

// Authors claimed that the AIE property of TPE could solve ACQ problem and give high PLQYs in aggregated state. But the aggregate of HQTPE only gave a low PLQY of ~0.1.

ANSWER: In the original version, we mistakenly assumed that **HQTPE** had a very poor aggregation in water, but neglected the presence of multiple hydroxyl groups. It is also mentioned in the suggestion of Reviewer#1 that PLQY of solid samples should be measured. (See Page 8 for new data. PLQY of solid **HQTPE** is 0.213.) The difference between the two data also suggests that dispersing **HQTPE** in poor solvents does not mean an entire aggregation. We hope this new data will address your concerns. Thanks for your advice.

// On the other hand, HQTPE can be easily oxidized and I do not think HQTPE is of high purity. Actually, the elemental analysis data of HQTPE are poor as shown in SI.

ANSWER: The new compounds were well characterized, and the NMR and MS data indicated that the new samples were of high purity, especially the NMR. It is true that **HQTPE** is extremely easy to be oxidized in the air, which may have undergone a little deterioration before elemental analysis, but this also shows the high oxygen affinity of this AIE core and its valuable application prospect from the side. Thanks for your reminding.

We have made new **HQTPE** sample and carried out the elemental analysis in time. The corrected data can be viewed on page 16.

// I think this work is actually routine and lacks novelty and significance, and there are no advantages of this molecule for pH and oxidant sensing.

ANSWER: We hope that the reviewer and editor can comprehensively consider our above reply and the newly revised version. This is the beginning of a series of our work that will lead to further research and more exciting applications. Thank you.

Reviewer #3:

In Page 6 line 108-110 the authors mentioned "During the oxidation process, newly generated QTPE has an amplification effect on fluorescence quenching, which accounts for the extremely high sensitivity of HQTPE." How to quantify the amplification on fluorescence quenching effect of newly generated QTPE?

ANSWER: By adjusting the ratio of reducing state and oxidation state in the system, we simulate the situation of continuous consumption of reducing state and continuous formation of oxidation state in the actual oxidation process. Without the amplification effect of **QTPE**, the fluorescence intensity of the system would show a linear relationship with the oxidation degree. In fact, the fluorescence in the system decreased sharply at the beginning of the oxidation process, and after the formation of some **QTPE**, the change of fluorescence intensity leveled off (no fluorescence at all). This suggests that when a **HQTPE** molecule is oxidized to **QTPE**, the fluorescence of this molecule will not only quench, but the oxidation product generated will weaken the fluorescence of other reducing species (**HQTPE**) in the system, which is the connotation of the so-called amplification effect on fluorescence.

Maybe it is our inadequate explanation that leads to your misunderstanding. We have added an extra explanation on this issue in page 6 to avoid similar confusion for readers. Thank you.

#The Figures 2A, 2D, 4A and 4D seem to be blurry.

ANSWER: Thanks for your correction. We have noticed the problem of image resolution in the original manuscript and hope it did not affect your review. Due to the large size of the original picture, the editing software automatically compressed the picture. We have urgently updated Figure 2 and 4 for review, and in the final submission we will submit a ZIP file containing all the pictures in high-definition. Thanks again for your reminding!

#There are lack of detailed information of S₁→S₀ transition and corresponding configuration interaction coefficients.

ANSWER: As NTO analysis used in our main article has already considered configuration interaction coefficients when producing the NTO orbitals showed, we add the contribution percentage of displayed NTO transitions to corresponding S₀-S₁ transitions on our figures (Figure 2 and Figure 4) in the main article. Meanwhile we have attached more data to the FMO analysis (under Figure S19 and S21) in SI. The newly added tables show the contribution percentage of transitions between noted FMOs to the S₀-S₁ transition, converted from the configuration interaction coefficients calculated through TD-DFT. The calculated contribution percentages are in agreement with our former analysis.

Moreover, we report the calculated configuration interaction coefficients in Table S3, in addition to many other calculated raw photophysical properties related to S₀-S₁ transition of corresponding molecules, including vertical excitation energy which is related to the emission wavelength, adiabatic excitation energy and oscillator strength.

With best regards.

Yours sincerely,

Yen Wei, Ph.D.

Chair Professor of Chemistry

Department of Chemistry, Tsinghua University, Beijing 100084, China

Tel: +86(10)158-01118976

E-mail: weiyen@tsinghua.edu.cn

Reviewers' comments:

Reviewer #1 (Remarks to the Author):

I am satisfied with the responses to the comments sent previously, so I recommend its publication in Communications Chemistry.

Reviewer #2 (Remarks to the Author):

Authors have addressed the comments and revised the paper accordingly. The paper may be acceptable for Communications Chemistry.

Reviewer #3 (Remarks to the Author):

The author gave a clear answer to comment 1 of commenter 3, but I personally do not quite agree with the author's conceptualization method, which is easy to mislead the understanding of the original concept of fluorescence quenching. I think this fluorescence quenching phenomenon cannot be enhanced. The reaction between the oxidation product and the reactant should be described independently.

Reviewer #4 (Remarks to the Author):

The authors have not properly addressed the comments of Reviewer #2, and there are still several questions need to be processed.

1. There is no doubt that the manuscript is a valuable research paper about a new intrinsic AIE-active core HQTPE with redox and pH responses. However, the investigations of the corresponding mechanisms are insufficient, and some more explanations about the related mechanisms should be added in the results and discussion section.
2. The AIE mechanism should be further studied by the method of SEM or TEM.
3. The authors should add some sentences involving the research progress of functionalized AIEgens in the introduction section.

Reviewer #1

I am satisfied with the responses to the comments sent previously, so I recommend its publication in Communications Chemistry.

NO ANSWER

Reviewer #2

// Authors have addressed the comments and revised the paper accordingly. The paper may be acceptable for Communications Chemistry.

NO ANSWER

Reviewer #3

The author gave a clear answer to comment 1 of commenter 3, but I personally do not quite agree with the author's conceptualization method, which is easy to mislead the understanding of the original concept of fluorescence quenching. I think this fluorescence quenching phenomenon cannot be enhanced. The reaction between the oxidation product and the reactant should be described independently.

ANSWER: Thank you very much for your reminding. When we read this part again following your idea, we found that there were indeed some expressions that might lead to misunderstanding in the original text. Our original meaning was that fluorescence attenuation was directly caused by the decrease in **HQTPE** concentration, and was amplified by the quenching effect of oxidation products, so that the detection will be more sensitive, but the quenching itself was not enhanced. In the new edition, the process of fluorescence attenuation is divided clearly into two parts: concentration decrease of **HQTPE** and quenching effect of **QTPE**. Specific changes are as follows:

(1) A control experiment on the linear relationship between the decrease of **HQTPE** concentration and the decay of fluorescence intensity has been added in Supplementary Information as Figure S14.

(2) The subtitle of "Oxidation-induced nonlinear quenching effect" has been changed to "Oxidation-induced intermolecular quenching effect". It adopts a unified expression with another work of our group on this similar phenomenon.

<https://doi.org/10.1002/adma.202004616>

Figure S9. Intermolecular quenching effect of TPEMI. (A) Fluorescence spectra of TPEMI and TPEMI-DA mixture in DMSO/water ($f_w = 90\%$) with different fraction of TPEMI from 0% (top) to 100% (bottom). The total concentration is 100 μM . (B) Plot of relative fluorescence intensity (I/I_0) at 475 nm versus fraction of TPEMI ($\lambda_{\text{ex}} = 320$ nm).

(3) Revise the relevant sentences and add new annotations to disambiguate in this section "Oxidation-induced nonlinear quenching effect".

Thanks again for your meticulous academic guidance and reminding. The previous edition was not well considered. We also hope these modifications can eliminate potential ambiguities and misunderstandings as much as possible.

Reviewer #4

There is no doubt that the manuscript is a valuable research paper about a new intrinsic AIE-active core HQTPE with redox and pH responses. However, the investigations of the corresponding mechanisms are insufficient, and some more explanations about the related mechanisms should be added in the results and discussion section.

ANSWER: Thank you for the advice. With more carefully study on the mechanisms, we find the faster nonradiative decay of the S_1 state to S_0 in **QTPE** and anion species of **HQTPE** is another factor contributing to the redox and pH responses of **HQTPE** in addition to the slower radiative decay process of them we have proposed. Detailed experiment evidence and theoretical reasoning according to the energy gap law have been added in the discussion section in manuscript.

The AIE mechanism should be further studied by the method of SEM or TEM.

ANSWER: Thanks for reminding. There is no doubt that electron microscopy is the most direct and powerful tool to prove whether it aggregates or not. In the new edition we have supplemented the scanning electron micrographs of **HQTPE** particles as Figure S12. It can be seen that **HQTPE** is completely dispersed in the organic solvent (THF), where is no fluorescence. In the water phase **HQTPE** aggregates into nanoparticles, where the fluorescence is strongest. This is completely consistent with the phenomenon of AIE, and can better demonstrate AIE mechanism. The relevant experiments has also been updated in the Experiment Section.

The authors should add some sentences involving the research progress of functionalized AIEgens in the introduction section.

ANSWER: Thanks for your advice and we re-read the introduction section. The content for the research progress of functionalized AIEgens was mostly focused on enumeration, without systematic summary and extension, which is indeed a deficiency in the original version. In the new edition, we add a systematic summary and comment on the previous works of functionalized AIEgens, making the introduction section full and complete. And the introduction from previous works to our work is added. Thanks again for your suggestions on this part.

With best regards.

Yours sincerely,

Yen Wei, Ph.D.

Chair Professor of Chemistry

Department of Chemistry, Tsinghua University, Beijing 100084, China

Tel: +86(10)158-01118976

E-mail: weiyen@tsinghua.edu.cn

REVIEWERS' COMMENTS:

Reviewer #3 (Remarks to the Author):

The author has made a good improvement on their manuscript. The current manuscript form should be acceptable to CC.

Reviewer #4 (Remarks to the Author):

This manuscript can be published in Communications Chemistry.

Reviewer #3

The author has made a good improvement on their manuscript. The current manuscript form should be acceptable to CC.

NO ANSWER

Reviewer #4

// This manuscript can be published in Communications Chemistry.

NO ANSWER

With best regards.

Yours sincerely,

Yen Wei, Ph.D.

Chair Professor of Chemistry

Department of Chemistry, Tsinghua University, Beijing 100084, China

Tel: +86(10)158-01118976

E-mail: weiyen@tsinghua.edu.cn